METHODS AND RESOURCES

# Versatile CRISPR/Cas9-mediated mosaic analysis by gRNA-induced crossing-over for unmodified genomes

**Sarah E. Allen**[1‡], **Gabriel T. Koreman**[1,2‡], **Ankita Sarkar**[1,2‡], **Bei Wang**[1,2], **Mariana F. Wolfner**[1]*, **Chun Han**[1,2]*

**1** Department of Molecular Biology and Genetics, Cornell University, Ithaca, New York, United States of America, **2** Weill Institute for Cell and Molecular Biology, Cornell University, Ithaca, New York, United States of America

‡ These authors contributed equally to this work, and are listed in alphabetical order.
* mariana.wolfner@cornell.edu (MFW); chun.han@cornell.edu (CH)

**Data Availability Statement:** All relevant data are within the paper and its Supporting Information files.

## Abstract

Mosaic animals have provided the platform for many fundamental discoveries in developmental biology, cell biology, and other fields. Techniques to produce mosaic animals by mitotic recombination have been extensively developed in *Drosophila melanogaster* but are less common for other laboratory organisms. Here, we report mosaic analysis by gRNA-induced crossing-over (MAGIC), a new technique for generating mosaic animals based on DNA double-strand breaks produced by CRISPR/Cas9. MAGIC efficiently produces mosaic clones in both somatic tissues and the germline of *Drosophila*. Further, by developing a MAGIC toolkit for 1 chromosome arm, we demonstrate the method's application in characterizing gene function in neural development and in generating fluorescently marked clones in wild-derived *Drosophila* strains. Eliminating the need to introduce recombinase-recognition sites into the genome, this simple and versatile system simplifies mosaic analysis in *Drosophila* and can in principle be applied in any organism that is compatible with CRISPR/Cas9.

## Introduction

Mosaic animals contain genetically distinct populations of cells that have arisen from 1 zygote. Mosaic animals have historically played important roles in the study of pleiotropic genes, developmental timing, cell lineage, neural wiring, and other complex biological processes. Given its genetic tractability, *Drosophila* has been a major system for generating and studying such mosaics [1], which have led to important discoveries such as developmental compartments [2], cell autonomy [3], and maternal effects of zygotic lethal genes [4]. Mosaic analysis is currently used to study tumor suppressors [5], signaling pathways [6], sleep–wake behaviors [7], cell fates [8], and neuronal lineages [9], among other biological processes.

The earliest mosaic analyses relied on spontaneous mitotic recombination [10], rare events in which a DNA double-strand break (DSB) during the $G_2$ phase of the cell cycle is repaired by homologous recombination, resulting in the reciprocal exchange of chromosomal arms

**Funding:** This work was supported by Cornell start-up funds and NIH grants (R01NS099125 and R21OD023824) awarded to C.H., by NIH grants (R01/R37-HD038921 and R03-HD101732) awarded to M.F.W., by an NIH training grant (T32-GM07273) awarded to the Cornell BMCB graduate program (A.P. Bretscher, P.I.) and that supported S.E.A. during part of this work, and by a Hunter R. Rawlings III Cornell Presidential Scholarship (RCPRS) awarded to G.K. The funders had no role in study design, data collection and analysis, decision to publish, or preparation of the manuscript. G.K., S.E.A., C.H., and M.F.W. received full or partial salary support from Cornell University; S.E.A., A.S., B.W., and C.H. received partial or full salary support from the NIH grants noted above.

**Competing interests:** The authors have declared that no competing interests exist.

**Abbreviations:** DFS, dominant female sterility; DGRP, Drosophila Genetic Reference Panel; DSB, double-strand break; Flp, Flippase; gRNA, guide RNA; HDR, homology-directed repair; HS, heat shock; MAGIC, mosaic analysis by gRNA-induced crossing-over; MARCM, mosaic analysis with a repressible cell marker; nBFP, nuclear blue fluorescent protein; NHEJ, nonhomologous end joining; nMAGIC, negative MAGIC; pMAGIC, positive MAGIC; Sec5, Secretory 5; Syx5, Syntaxin 5.

between homologous chromosomes distal to the site of the DNA crossover (reviewed in [11]). Ionizing radiation, such as X-rays [12], cause DSBs and thus were later used in *Drosophila* to increase the baseline level of mitotic recombination [13]. However, ionizing radiation breaks genomic DNA at random locations and is associated with a high degree of lethality.

To overcome these limitations, the yeast Flippase (Flp)/FRT system was introduced into *Drosophila* to mediate site-specific recombination at FRT sites [14,15], enabling the development of an ever-expanding toolbox with enhanced power and flexibility for mosaic analysis [15–19]. This system requires that both homologous chromosomes carry FRT sites at the same position proximal (relative to the centromere) to the gene of interest, and an independent marker on one of the homologs to allow visualization of the genetically distinct clones [15]. For mosaic analysis in the *Drosophila* germline, a dominant female sterility (DFS) *ovo*$^{D1}$ transgene was combined with Flp/FRT methods, allowing production of, and selection for, germline clones homozygous for a mutation of interest in a heterozygous mother [20,21]. In this "Flp-DFS" technique, egg production from *ovo*$^{D1}$-containing heterozygous and homozygous germline cells is blocked, resulting in progeny derived exclusively from germline clones lacking *ovo*$^{D1}$ that were generated by mitotic recombination [22]. Mosaic analysis based on somatic recombination has also been achieved in mice using the Cre-LoxP system and the reconstitution of fluorescent protein genes as markers [23,24]. Despite these successes, site-specific recombination systems have not been widely used for mitotic recombination in model animals beyond *Drosophila melanogaster* due to the challenging task of introducing recombinase-recognition sites into centromere-proximal regions for every chromosome.

Given the power of mosaic animals in biological research, it would be useful to have a more general approach for inducing interhomolog mitotic recombination in any organism, circumventing the challenges just mentioned. The CRISPR/Cas9 system has great potential for extending mosaic analysis, because it can create targeted DSBs in the genomic DNA of a wide array of organisms [25]. This binary system requires only the Cas9 endonuclease and a guide RNA (gRNA) that specifies the DNA target site [26], both of which can be introduced into the cell independently of the location of the target site. CRISPR/Cas9-induced DSBs can be repaired either by nonhomologous end joining (NHEJ) or homology-directed repair (HDR). So far, most CRISPR/Cas9 applications in animals have been focused on NHEJ-mediated mutagenesis and HDR-mediated gene replacement [27]. Recently, several studies demonstrated that CRISPR/Cas9-induced DSBs can also induce targeted mitotic recombination in yeast and in the germlines of *Drosophila*, houseflies, and tomatoes [28–31], suggesting the possibility of exploiting this property of CRISPR/Cas9 for mosaic analysis. Here, we report <u>m</u>osaic <u>a</u>nalysis by <u>g</u>RNA-<u>i</u>nduced <u>c</u>rossing-over (MAGIC), a novel technique for mosaic analysis based on CRISPR/Cas9. This method can be used to generate mosaic clones in both the *Drosophila* soma and germline. Based on this method, we built a convenient toolkit to generate and label mosaic clones for genes located on chromosome arm 2L. We demonstrate the success of our toolkit for mosaic analysis in the soma and the germline and show its applications in analyzing gene functions in neuronal dendrite development. Lastly, we also demonstrate that MAGIC can be used successfully with unmarked wild-derived strains, indicating that this method can in principle be extended to organisms for which recombinase-based toolkits are not yet available.

## Results

### Rationale for MAGIC

MAGIC relies on the action of gRNA/Cas9 in a proliferating cell during G$_2$ phase to generate a DSB at a specific position on 1 chromatid of a homologous pair (Fig 1A). The DSB can induce

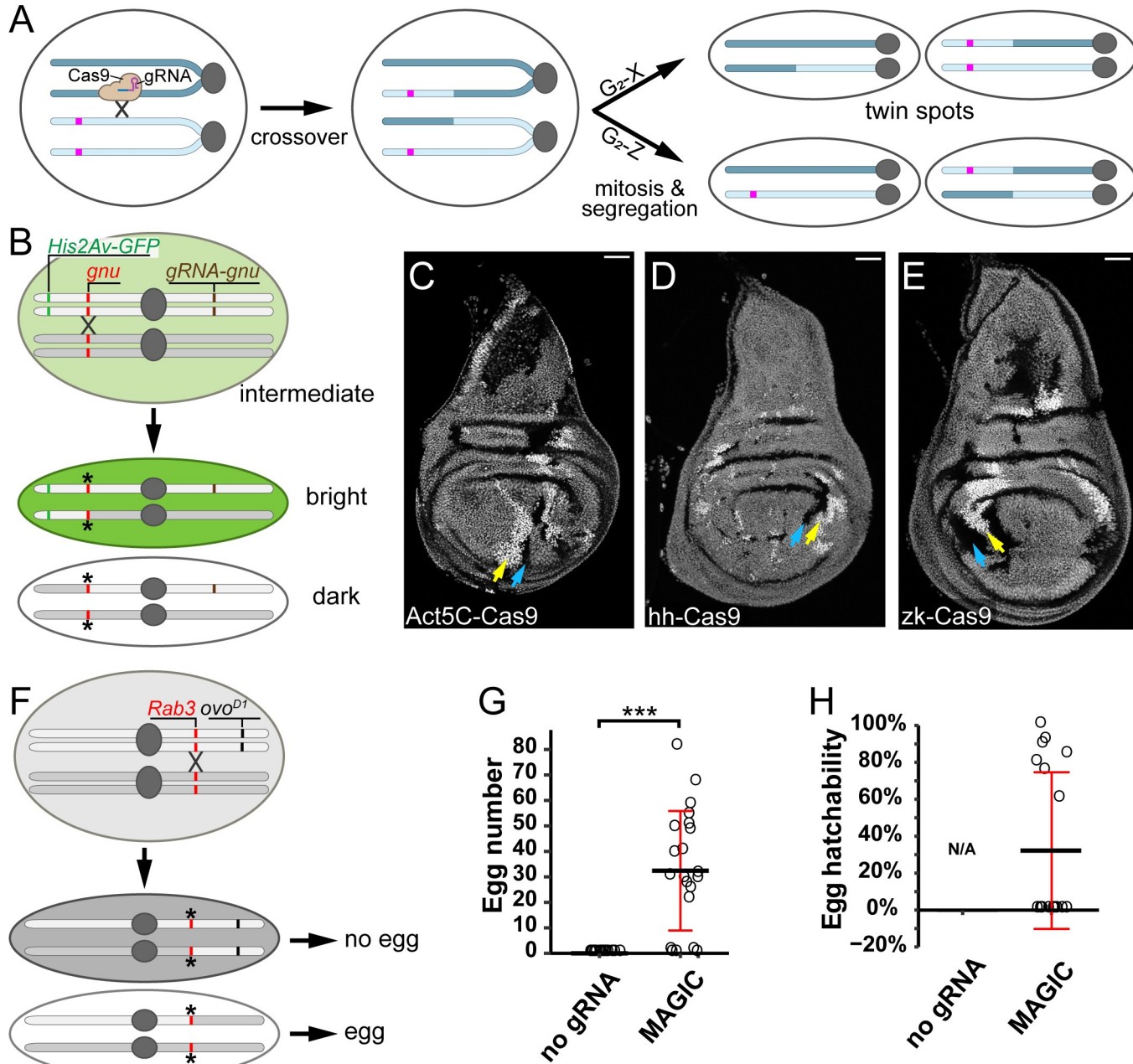

**Fig 1. CRISPR-induced crossover generates somatic and germline clones in *Drosophila*.** (A) A diagram illustrating the principle of MAGIC. The magenta bar indicates a marker that is lost from 1 daughter cell and becomes homozygous in the other in $G_2$-X-generated twin spots. (B) Strategy for generating mosaic clones using the *His2Av-GFP* marker and a crossover at the *gnu* locus. Locations of the GFP marker *(His2Av-GFP*, green bar), the gRNA (*gnu-g*RNA, brown bar), and the target of the gRNA (*gnu*, red bar) are shown. Asterisks indicate mutated gRNA target sites. (C–E) Mosaic clones in wing discs visualized by levels of *His2Av-GFP* expression, as described in the text. A pair of arrows indicate *His2Av-GFP*$^{+/+}$ (yellow) and *His2Av-GFP*$^{-/-}$ (blue) cells in each panel. Clones were induced by *Act5C-Cas9* (C), *hh-Cas9* (D), and *zk-Cas9* (E). Scale bars, 50 μm. (F) Strategy for generating and detecting clones in the female germline using *ovo*$^{D1}$. Figure labels are analogous to those in 1B; the gRNA is on another chromosome. (G) Number of eggs produced by females carrying *nos-Cas9* and *ovo*$^{D1}$, with (MAGIC) or without (no gRNA) *gRNA-Rab3*. $^{***}p < 0.001$, Student *t* test. *n* = number females: no gRNA (*n* = 18); MAGIC (*n* = 18). (H) Hatchability of eggs produced females carrying *nos-Cas9* and *ovo*$^{D1}$, with (MAGIC) or without (no gRNA) *gRNA-Rab3*. *n* = number females: no gRNA (N/A); MAGIC (*n* = 18). For quantifications in G and H, black bar, mean; red bar, SD. The data underlying this Figure can be found in S1 Data. GFP, green fluorescent protein; gRNA, guide RNA; MAGIC, mosaic analysis by gRNA-induced crossing-over.

a crossover between this chromatid and a chromatid from the homologous chromosome, resulting in an exchange of chromosome segments between the 2 chromatids at the location of the DSB. During the subsequent mitotic segregation of chromosomes, recombinant chromatids sort into different daughter cells in a $G_2$-X segregation event, generating "twin spots," which contain 2 genetically distinct populations of cells homozygous for the chromosome segment distal to the exchange [32]. Alternatively, recombinant chromatids can sort into the same daughter cell in $G_2$-Z segregation, resulting in 2 daughter cells that still maintain heterozygocity at every locus [32]. By introducing a genetic marker distal to the gRNA target site in one of the homologous chromosomes, it is possible to distinguish homozygous twin spots among heterozygous cells based on the gain of copies of and/or loss of the genetic marker.

## Using CRISPR-induced crossover to generate clones in the *Drosophila* soma and germline

For our initial tests of the ability of MAGIC to generate mosaic clones in somatic tissues, we used ubiquitously expressed gRNAs to induce DSBs at the *gnu* locus and a ubiquitous fluorescent marker, *His2Av-GFP* [33], to trace clones (Fig 1B). Both *His2Av-GFP* and *gnu* are located on the left arm of chromosome 3 (3L), and *His2Av-GFP* is distal to *gnu*. We chose *gnu* as our gRNA target because we had already made an efficient *gRNA-gnu* line for other purposes (to be published elsewhere); furthermore, this gene is only required maternally for embryonic development [34], so mutations of *gnu* are not expected to affect the viability or growth of somatic cells. We induced clones using 3 different Cas9 transgenes, each of which is expressed in the developing wing disc under the control of a different enhancer. With all 3 Cas9s, we observed twin spots consisting of bright *His2Av-GFP* homozygous clones abutting GFP-negative clones in the midst of *His2Av-GFP/+* heterozygous cells (Fig 1C–1E) in every imaginal disc examined, demonstrating the feasibility of MAGIC for generating somatic mosaics.

Given that CRISPR/Cas9 is active in both the soma and the germline of *Drosophila* [35–37], we next tested for MAGIC clone induction in the germline by using the DFS technique and the germline-specific *nos-Cas9* [36] (S1A Fig). We used an *ovo*[D1] transgene located on chromosome arm 2R and induced DSBs at the *Rab3* locus, which is located on the same arm proximal to the location of *ovo*[D1] (Fig 1F). Since *Rab3* is a nonessential gene expressed only in neurons [38], its disruption in the female germline should affect neither egg production nor embryonic development of the progeny. Due to the dominant effect of *ovo*[D1], restoration of egg production can only result from mitotic recombination proximal to *ovo*[D1] (e.g., at *Rab3* in this case), followed by generation of *ovo*[D1]-negative clones. As expected, control females that contained *ovo*[D1] and *nos-Cas9*, but not *gRNA-Rab3*, did not produce any eggs. In contrast, most females carrying all 3 components produced 20 to 90 eggs each (Fig 1G), many of which hatched into larvae (Fig 1H), suggesting successful mitotic recombination.

The results above together show that, like the Flp/FRT system, MAGIC is an effective approach for generating homozygous clones via mitotic recombination in both *Drosophila* soma and germline, consistent with the high frequency of CRISPR-induced exchange of chromosomal arms previously demonstrated in the *Drosophila* germline [28].

## A toolkit for generating labeled clones for genes on chromosome arm 2L

Towards making MAGIC a general approach for analyzing *Drosophila* genes, we built a toolkit for genes located on chromosome arm 2L as a proof of principle. We designed transgenic constructs that each integrate 2 features simultaneously: ubiquitously expressed gRNAs that target a pericentromeric region and a ubiquitously expressed marker for visualizing clones. The constructs were inserted into a distal position of 2L. When a gRNA-marker-bearing chromosome

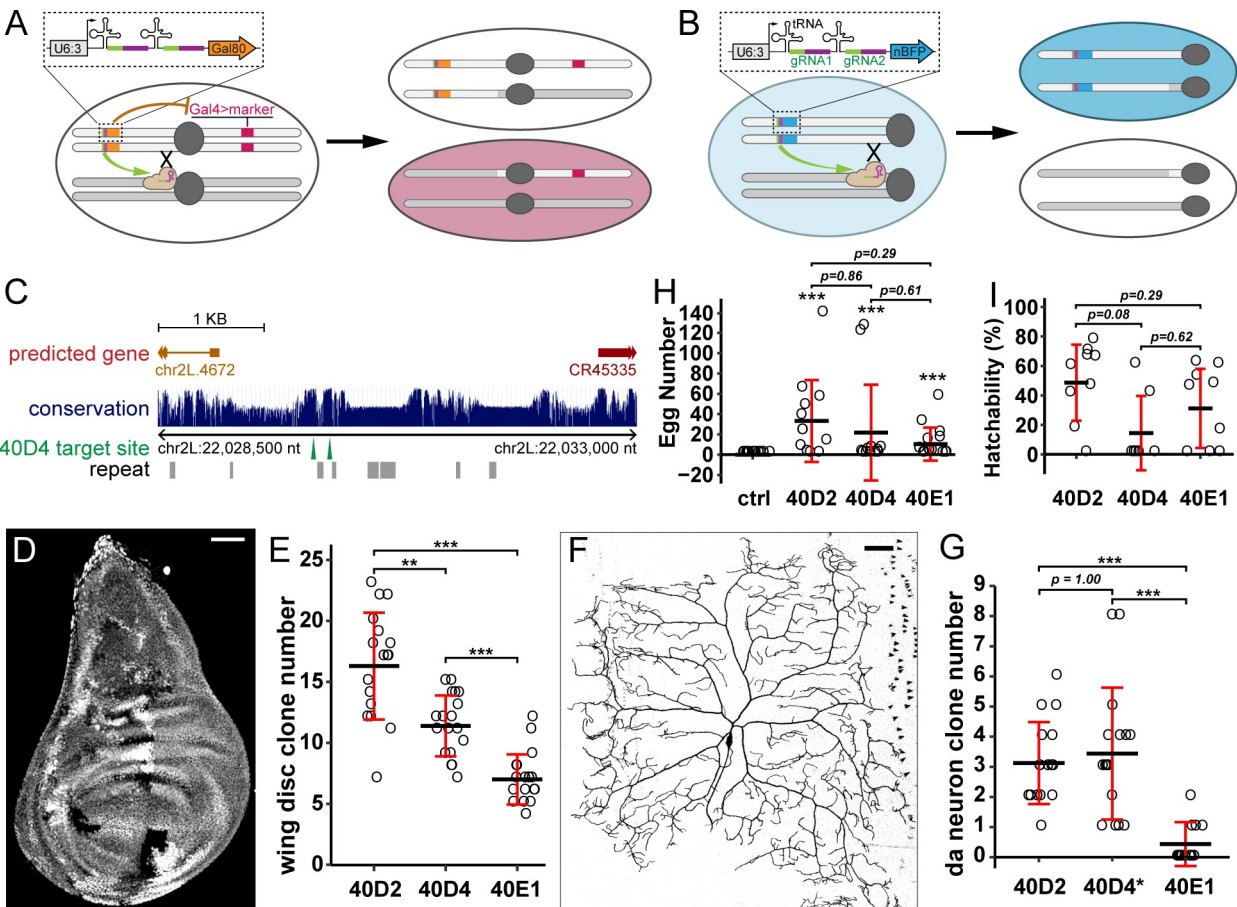

**Fig 2. A toolkit for generating labeled clones for genes on chromosome arm 2L.** (A) Strategy of pMAGIC for generating positively labeled clones for genes on 2L using Gal80. (B) Strategy of nMAGIC for genes on 2L using nuclear BFP marker. (C) A map for gRNA-(40D4) target sites derived from the UCSC Genome Browser. The height of the blue bar corresponds to the level of conservation among 23 *Drosophila* species. The locations of predicted genes (thick line, exon or mature RNA; thin line, intron; arrow, gene orientation), repeated DNA sequences (gray boxes), gRNA target sites (green arrowheads), and chromosomal coordinates are indicated. (D) A wing imaginal disc showing twin-spot clones generated by gRNA-40D2(nBFP) paired with *hh-Cas9*. (E) Number of clones in wing discs using 3 different gRNA(nBFP) constructs. *n* = number of discs: 40D2 (*n* = 17); 40D4 (*n* = 18); 40E1 (*n* = 18). **$p \leq 0.01$, ***$p \leq 0.001$; Welch ANOVA and Welch *t* tests, *p*-values corrected using Bonferroni method. (F) A C4da neuron clone generated by gRNA-40D2(Gal80) paired with *SOP-Cas9* and *ppk>CD4-tdTom*. (G) Quantification of C4da clones using 3 different gRNA(Gal80) constructs. *n* = number of larvae: 40D2 (*n* = 16); 40D4 (*n* = 16); 40E1 (*n* = 16). Clones were visualized by *ppk>CD4-tdTom*. ***$p \leq 0.001$; Welch ANOVA and Welch *t* tests, *p*-values corrected using the Bonferroni method. The asterisk on 40D4 indicates frequent labeling of C3da clones. (H) Number of eggs produced by using *ovo^{D1}*(2L), *nos-Cas9*, and 3 different gRNA(nBFP) constructs. The ctrl has no gRNA. *n* = number of females: ctrl (*n* = 13); 40D2 (*n* = 12); 40D4 (*n* = 12); 40E1 (*n* = 14). Asterisks are from post hoc single sample *t* tests that compare the EMM of each gRNA against the control (μ = 0); ***$p \leq 0.001$. *p*-Values are from contrasts of EMMs of each gRNA (excluding control). EMMs are based on a negative binomial model. All *p*-values were corrected using the Tukey method. (I) Hatchability of eggs produced by females carrying *ovo^{D1}*(2L), *nos-Cas9*, and gRNA(nBFP) constructs. *n* = number of females: 40D2 (*n* = 9); 40D4 (*n* = 7); 40E1(*n* = 9). *p*-Values are from contrasts of EMMs of proportions of hatched/nonhatched eggs for each gRNA based on a binomial, mixed-effects model and were corrected using the Tukey method. For E, G, and H: black bar, mean; red bar, SD. Scale bars, 50 μm. The data underlying this Figure can be found in S1 Data. BFP, blue fluorescent protein; ctrl, control; EMM, estimated marginal mean; gRNA, guide RNA; nBFP, nuclear blue fluorescent protein; nMAGIC, negative MAGIC; pMAGIC, positive MAGIC.

is used together with an unmodified second chromosome, clones that are homozygous for nearly the entirety of the unmodified arm can be visualized by the loss of the marker. Positive labeling with pMAGIC utilizes a Gal80 marker [18], which suppresses Gal4-driven expression of a fluorescent reporter (Fig 2A). Therefore, only the cells that lose the Gal80 transgene will be fluorescently labeled, similarly to mosaic analysis with a repressible cell marker (MARCM) [18]. In contrast, negative MAGIC (nMAGIC) expresses a nuclear blue fluorescent protein

(nBFP) reporter, such that clones homozygous for the unmodified arm are marked by the loss of nBFP expression (Fig 2B).

To identify appropriate gRNA target sites, we surveyed the pericentromeric sequences of 2L for sequences that met 3 criteria: (1) being reasonably conserved so that DSBs can be induced in most *Drosophila* strains; (2) not functionally critical and being distant from essential sequences so that indel mutations in nearby regions would not disrupt important biological processes; and (3) unique in the genome and predicted to have a low likelihood of off-targeting. Therefore, for each MAGIC construct, we chose a pair of nonrepeat gRNA target sequences in an intergenic region to enhance the chance of DSBs. The 2 gRNA target sequences are closely linked to reduce the risk of large deletions (Fig 2C). In addition, we preferentially chose sequences that are conserved among closely related *Drosophila* species (*D. melanogaster*, *D. simulans*, and *D. sechellia*) but not in more distant species. Considering the varying efficiencies of different gRNA target sequences, we selected 3 pairs of gRNAs targeting 3 chromosomal locations (40D2, 40D4, and 40E1) and tested their ability to produce clones in wing discs, neurons, and the germline.

Clones were induced in a specific tissue by a Cas9 transgene that is expressed in precursor cells of that tissue. We used *hh-Cas9* [39] for nMAGIC in the wing imaginal disc (Fig 2D), *SOP-Cas9* [39] for pMAGIC in larval class IV dendritic arborization (C4da) sensory neurons (Fig 2F), and *nos-Cas9* for the female germline (S1B Fig). gRNAs targeting 40D2 consistently performed the best in generating clones in wing discs and C4da neurons (Fig 2E and 2G) and appear to be the most efficient in the germline, even though the differences in the germline were not statistically significant (Fig 2H and 2I). Although the overall efficiencies of *gRNA-40D2* and *gRNA-40D4* in inducing clones in da neurons are similar (Fig 2G and S2A Fig), *gRNA-40D4* induced more clones in a different type of neuron (class III). These results indicate that we have created an efficient MAGIC toolkit for genes located on chromosome arm 2L. Analogous toolkits will be made for other chromosome arms, using the same methods, and will be reported separately once they are generated. A list of potential gRNA target sites for those chromosome arms is in S2 Table.

## Inducible clone generation and comparison with Flp/FRT-mediated mosaic analysis

Existing Flp/FRT-recombination systems allow temporal control of clone induction through the use of Flp transgenes driven by heat shock (HS) promoters [40,41], providing great flexibility to mosaic analysis. To test if gRNA-induced crossing-over can generate clones via similar approaches, we paired nMAGIC line *gRNA-40D2(nBFP)* with a HS-inducible Cas9 that was reported recently [42]. A single 1-hour HS at 37°C of early third instar larvae robustly induced clones in the wing imaginal disc (Fig 3B and 3C), whereas larvae of the same genotype that did not experience HS showed nearly no clones in the wing disc (Fig 3A and 3C). These results show that MAGIC is compatible with inducible Cas9.

As gRNA-induced DSBs and Flp/FRT yield interchromosomal exchange through distinct mechanisms, we wondered if MAGIC and Flp/FRT exhibit different efficiencies of clone induction. To directly compare these 2 systems in the wing imaginal disc, we chose 2 pairs of Cas9 and Flp transgenes, with each pair controlled by an identical enhancer (*zk* or *hh*). FRT-mediated recombination was induced between two *FRT^40A*-bearing chromosomes that carry ubiquitously expressed nuclear GFP (nGFP) or nuclear RFP (nRFP) (Fig 3G). *FRT^40A* was selected as the FRT site for comparison because of the availability of 2 different fluorescent markers on *FRT^40A* arms, which makes the measurement of clone numbers and sizes convenient. gRNA-induced crossing-over occurred between *gRNA-40D2(nBFP)* and *nRFP FRT^40A*

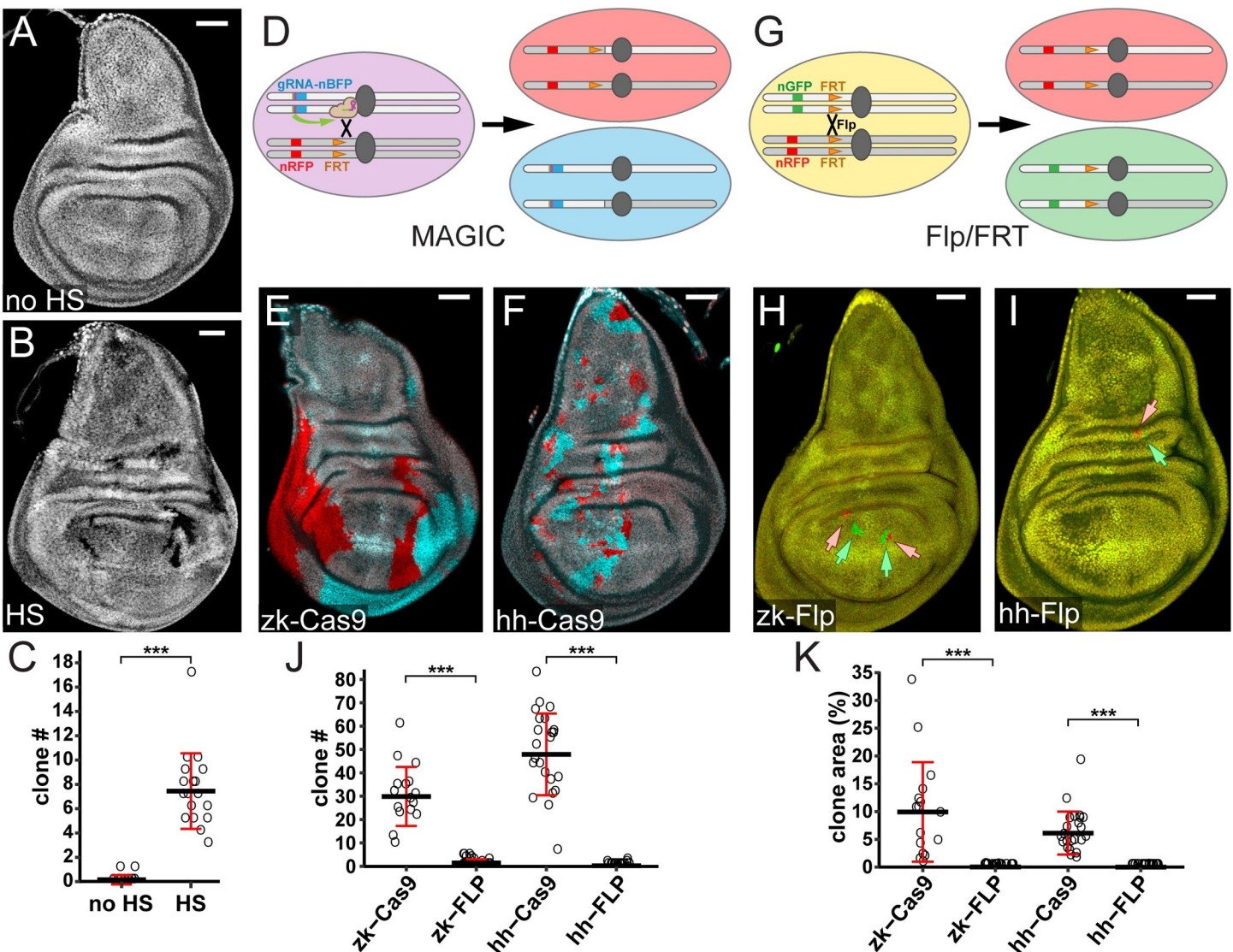

**Fig 3. Inducible MAGIC and comparison with Flp/FRT-mediated mosaic analysis.** (A and B) Wing imaginal discs of animals with *gRNA-40D2(BFP)* and HS-Cas9 with (B) and without (A) HS at 72 h at egg laying. (C) Quantification of clone number per disc induced by HS. *n* = number of discs: no HS (*n* = 13), HS (*n* = 18). ***$p \leq 0.001$, Welch 2-sample *t* test. (D) Diagram of twin spot labeling using *gRNA-40D2(nBFP)* and *nRFP FRT^{40A}* chromosomes via MAGIC. (E and F) Wing imaginal discs showing clones generated by MAGIC using *zk-Cas9* (E) and *hh-Cas9* (F) visualized with nBFP (cyan) and nRFP (red). (G) Diagram of twin spot labeling using *nRFP FRT^{40A}* and *nGFP FRT^{40A}* chromosomes through Flp/FRT-mediated site-specific recombination. (H and I) Wing imaginal discs showing clones generated by FRT/Flp-mediated site-specific recombination using *zk-Flp* (H) and *hh-Flp* (I) and visualized with nGFP (green, indicated by green arrows) and nRFP (red, indicated by pink arrows). (J and K) Quantification of clone number per disc (J) and percentage of total clone area (K). *n* = number of discs: *zk-Cas9* (*n* = 16), *zk-FLP* (*n* = 18), *hh-Cas9* (*n* = 23), *hh-FLP* (*n* = 19). ***$p \leq 0.001$, Welch 2-sample *t* test. Scale bars, 50 μm. The data underlying this Figure can be found in S1 Data. Flp, Flippase; FRT, flippase recognition target; gRNA, guide RNA; HS, heat shock; MAGIC, mosaic analysis by gRNA-induced crossing-over; nBFP, nuclear blue fluorescent protein; nGFP, nuclear green fluorescent protein; nRFP, nuclear red fluorescent protein.

chromosomes (Fig 3D). As expected, both *zk-Cas9* and *hh-Cas9* generated clones robustly in the wing disc, with *zk-Cas9* causing fewer but larger clones and *hh-Cas9* causing more but smaller clones (Fig 3E, 3F, 3J, and 3K). Surprisingly, very few small clones were observed in wing discs when the *FRT^{40A}* chromosomes were paired with either Flp transgene (Fig 3H–3K). Because clone frequencies were low with both Flp lines, we did not detect statistical differences between them. Given that the low efficiency of Flp/FRT in these experiments may be due to

the specific site of $FRT^{40A}$, these results may not indicate a general difference in efficiency between MAGIC and Flp/FRT. Nevertheless, our results show that MAGIC can be reliably and efficiently used for mosaic analysis.

## Mosaic analysis of neuronal dendrite development

To evaluate the utility of our MAGIC toolkit for characterizing gene function at the single-cell level, we combined the pMAGIC line *gRNA-40D2(Gal80)* with mutations on 2L that affect dendrite morphogenesis in C4da neurons by disrupting vesicular trafficking. We first used 2 genes, *Secretory 5* (*Sec5*) [43] and *Rab5* [44], which have been shown to be required for dendrite growth. We observed marked dendrite reduction in C4da clones carrying homozygous mutations in these genes (Fig 4A, 4B, and 4D), recapitulating previously published results using MARCM with the same mutants [43,44]. A third gene, *Syntaxin 5* (*Syx5*), was identified in our unpublished RNA interference (RNAi) screens. Clones carrying a null mutation of *Syx5* produced the most dramatic dendrite reduction, with almost all terminal dendrites eliminated (Fig 4C–4E), consistent with the expected role of Syx5 in the endoplasmic reticulum to Golgi vesicle trafficking [45]. Therefore, our MAGIC reagents for 2L can be used to characterize gene functions in single cells with a power analogous to that of MARCM but with a much simpler system.

## Generation of clones by MAGIC in fly lines with wild-derived genomes

An advantage of MAGIC compared to Flp/FRT-based mitotic recombination systems is that MAGIC does not require that FRT sites be present on the homologous chromosomes to mediate crossing-over. MAGIC is therefore easier to apply to fly strains with wild-derived genomes and even potentially to organisms in which genetic systems like Flp/FRT are not routinely available. To test the applicability of MAGIC to unmarked strains with wild-derived genomes, we crossed *gRNA-40D2(nBFP); hh-Cas9* to 5 randomly chosen lines from the *Drosophila* Genetic Reference Panel (DGRP) [46], a set of unique strains established from flies captured in the wild. In all cases, we observed efficient clone induction in wing imaginal discs (Fig 5A–5E), demonstrating the potential of MAGIC for allowing future mosaic analysis of the function of natural alleles residing on wild-derived chromosomes.

## Discussion

We present here a new technique that we named MAGIC (mosaic analysis by gRNA-induced crossing-over) for mosaic analysis based on CRISPR-induced mitotic recombination. We

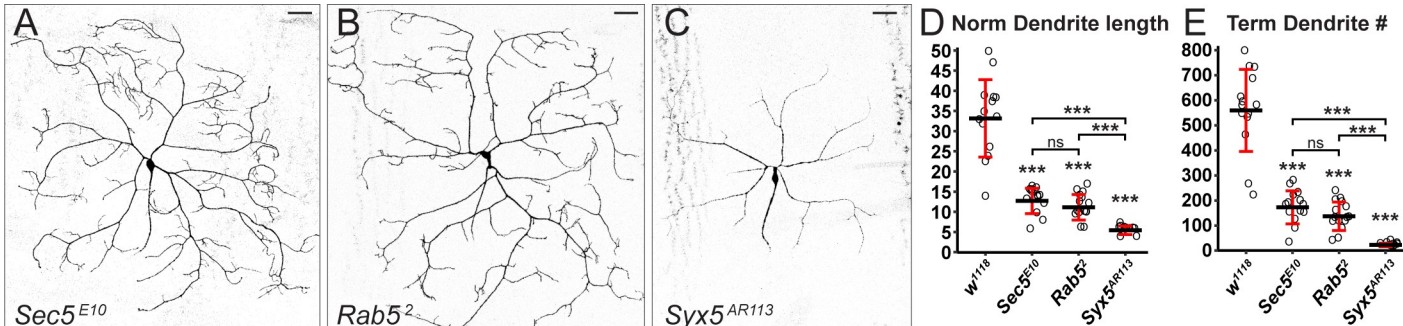

**Fig 4. Mosaic analysis of example genes in neuronal dendrite development.** (A–C) MAGIC clones of C4da neurons mutant for $Sec5^{E10}$ (A), $Rab5^{2}$ (B), and $Syx5^{AR113}$ (C), visualized by *ppk>MApHS*. Scale bars, 50 μm. (D and E) Quantification of normalized dendrite length (total dendrite length/segment width) (D) and terminal branch number (E) of C4da clones of wild-type control ($w^{1118}$) and mutants. *n* = number neurons: $w^{1118}$ (*n* = 14); $Sec5^{E10}$ (*n* = 14); $Rab5^{2}$ (*n* = 15); $Syx5^{AR113}$ (*n* = 10). The asterisk above each genotype indicates comparison with the control. $^{***}p \leq 0.001$, ns, not significant, Welch ANOVA and Welch *t* tests, *p*-values corrected using the Bonferroni method. The data underlying this Figure can be found in S1 Data. MAGIC, mosaic analysis by gRNA-induced crossing-over.

**Fig 5. MAGIC generates clones with DGRP lines.** (A–E) Clones in wing imaginal discs by pairing *gRNA-40D2(BFP); hh-Cas9* with DGRP line 109 (A), 223 (B), 237 (C), 280 (D), and 356 (E). Scale bars, 50 μm. DGRP, *Drosophila* Genetic Reference Panel; MAGIC, mosaic analysis by gRNA-induced crossing-over.

show that MAGIC is capable of efficiently producing mosaic tissues in both the *Drosophila* soma and germline, using gRNAs targeting various chromosomal locations. Integrated gRNA-marker constructs enable visualization of homozygous clones by the presence or absence of fluorescent reporters. As demonstrated by our 2L toolkit, MAGIC is simple and effective to use. Similar MAGIC reagents will be generated for all other chromosome arms to allow for genome-wide characterization of gene functions.

Existing techniques for mosaic analysis have led to many major advances and findings in cell and developmental biology [2–4]. MAGIC provides additional flexibility, in that it does not require any modifications of the test chromosome, such as recombination with an existing FRT site. Integrating gRNAs and genetic markers into 1 transgenic construct also reduces the number of necessary genetic components. As a result, gRNA-marker transgenes can be combined with existing mutant libraries to perform MAGIC with very little additional effort. Moreover, with MAGIC, the site of mitotic recombination is limited only by the gRNAs that can be designed. Therefore, MAGIC can in principle permit mosaic analysis of mutations that were previously very difficult or impossible to study with existing techniques, including those near centromeres and the ones associated with transgenic constructs that contain FRT sequences [47].

Successful use of MAGIC requires 4 considerations. First, gRNAs should be carefully designed to eliminate, or minimize, potential off-target effects. Off-target prediction algorithms based on empirical evidence [48–51], which are constantly being improved, should facilitate this process, even though the possibility of off-targeting cannot be completely excluded. Second, our results suggest that the gRNA target sequence strongly influences the efficiency of clone induction, likely by affecting the frequency of DNA DSBs in premitotic cells. Therefore, for mosaic analysis of a specific chromosomal arm, it is beneficial to compare a few candidate gRNA targets and select the most effective one. Third, because perfect DSB repair will recreate the gRNA target site and allow for one more round of Cas9 cutting, most cells that have expressed Cas9 in their lineages are expected to eventually harbor indel mutations that disrupt the gRNA target site, regardless of whether or not the DSBs have led to mitotic recombination. However, this caveat can be mitigated by choosing gRNA sites in non-critical sequences, which can be validated by crossing gRNA lines to a ubiquitous Cas9 or by comparing gRNA-induced control clones to wild-type cells. To facilitate selection of appropriate gRNA target sites, we list 3 sets of potential gRNA target sequences for each major chromosome arm in S2 Table. Finally, since only DNA DSBs in the $G_2$ phase can lead to clone generation, the timing of Cas9 action is expected to be critical for MAGIC. For the cell type in

question, an ideal Cas9 should be expressed in the precursor cells, as too early expression can mutate gRNA target sites prematurely and too late expression will lead to unproductive DSBs.

Perhaps the most exciting aspect of MAGIC is its potential for use with wild-derived *Drosophila* strains and in organisms beyond *Drosophila*. DGRP and other wild-derived strains have played important roles in identification of natural alleles that are associated with certain phenotypic variations [52–55]. However, it has been difficult to investigate the effect of homozygosity for many alleles within these strains using available genetic tools (e.g., Flp/FRT, Gal4 drivers, and fluorescent markers) for technical reasons. For example, recombining existing FRT sites into DGRP chromosomes without removing potentially interacting genetic variants present in these stocks will be very challenging. By crossing MAGIC lines carrying both gRNA-marker and Cas9 to DGRP strains, one can validate causal effects of specific natural alleles in cellular or developmental processes in a natural-genome context and a tissue-specific manner. Similarly, the DGRP can also be used in MAGIC-based genetic screens to identify natural alleles that, when made homozygous, can cause or modify certain phenotypes. For these applications, germline-expressing Cas9 lines should be avoided to prevent heritable mutations of gRNA target sites.

Importantly, MAGIC can, in theory, be utilized in a wide array of organisms that are compatible with CRISPR/Cas9 [56]. In model systems that allow for transgenesis of gRNA-marker constructs, such as mouse, zebrafish, and *Xenopus*, Cas9 can be introduced by injection or virus transduction to further simplify genetic manipulations. It is worth noting that *Drosophila* is unusual in that its homologous chromosomes pair during mitosis, which facilitates mitotic recombination [57], and that its recombinant chromatids resulting from interchromosomal exchange in $G_2$ phase predominantly undergo X segregation [32]. Although other organisms may not share these properties, homology-directed recombination in mitotic cells has been documented in other organisms such as *Saccharomyces cerevisiae* and mammals [58–60]. Therefore, the flexibility and power of mosaic analysis that are familiar to the *Drosophila* research community will likely be in reach of researchers who study organisms which have not, or have rarely, been amenable to mosaic analysis.

## Materials and methods

### Fly stocks and husbandry

See the Key Resource Table (S1 Table) for details of fly stocks used in this study. Most fly lines were either generated in the Han and Wolfner labs or obtained from the Bloomington *Drosophila* Stock Center or the *Drosophila* Genetic Reference Panel [46]. *HS-Cas9* [42] was a gift from Tuzmin Lee. *hh-Flp* was generated by converting *R28E04-Gal4^{attP2}* into a Flp-expressing line using the homology assisted CRISPR knock-in (HACK) method [61]. The details of the conversion will be reported elsewhere. All flies were grown on standard yeast-glucose medium, in a 12:12 light/dark cycle, at room temperature (22 ± 1°C, for the egg laying assay) or 25°C (for larval assays) unless otherwise noted. Virgin males and females for mating experiments were aged for 3 to 5 days. Virgin females were aged on yeasted food for germline mosaic analysis.

To test germline clone induction, we combined *nos-Cas9* and *ovo^{D1}*, and then the gRNA in 2 sequential crosses in schemes shown in S1 Fig.

To visualize clones of C4da neurons, we used *ppk-Gal4 UAS-CD4-tdTom* [67,68] (Fig 2) and *ppk-Gal4 UAS-MApHS* [69] (Fig 3, only the tdTom channel is shown).

To temporally induce clones in the wing disc, larvae (72 hours after egg laying) of *w; gRNA-40D2(nBFP)/+; HS-Cas9/+* were heat shocked for 1 hour at 37°C, and their discs were examined at 96 hours after egg laying.

## Molecular cloning

*zk-Cas9*: The entry vector pENTR221-ZK2 [62] and the destination vector pDEST-APIC-Cas9 (Addgene 121657) were combined in a Gateway LR reaction to generate the expression vector pAPIC2-ZK2-Cas9.

*MAGIC gRNA-marker vectors*: gRNA-marker vectors were constructed similarly to pAC-U63-tgRNA-Rev (Addgene 112811, [39]) but have either a ubi-nBFP (in pAC-U63-tgRNA-nlsBFP) or a ubi-Gal80 (in pAC-U63-tgRNA-Gal80) marker immediately after the U6 3′ flanking sequence. The markers contain a Ubi-p63E promoter, mTagBFP-NLS or Gal80 coding sequence, and His2Av polyA sequence. The Ubi-p63E promoter was amplified from *Ubi-CasExpress* genomic DNA using the oligonucleotides TTAATGCGTATGCAT TCTAGTggccatggcttgctgttcttcgcgttc and TTGGATTATTctgcgggcagaaaatagagatgtggaaaattag. mTagBFP-NLS was synthesized as a gBlock DNA fragment (Integrated DNA Technologies, Coralville, Iowa 52241, United States of America). Gal80 coding sequence was PCR amplified from pBPGAL80Uw-4 (Addgene 26235) using the oligonucleotides aaaaaaaaatcaaaATGAGC GGTACCGATTACAACAAAAGGAGTAGTGTGAG and GCCGACTGGCTTAGTTAatta attctagaTTAAAGCGAGTAGTGGGAGATGTTG. The His2Av polyA sequence was PCR amplified from pDEST-APLO (Addgene 112805). DNA fragments were assembled together using NEBuilder DNA Assembly (New England Biolabs, Ipswich, Massachusetts 01938, USA).

*gRNA expression vectors*: For *gnu* and *Rab3*, gRNA target sequences were cloned into pAC-U63-tgRNA-Rev as described [39]. For gRNAs targeting 2L, gRNA target sequences were cloned into pAC-U63-tgRNA-nlsBFP and pAC-U63-tgRNA-Gal80 using NEBuilder DNA Assembly. In the gRNA-marker constructs, the tRNA between the first and second gRNAs is a *Drosophila* glutamine tRNA (cagcgcgcGGTTCCATGGTGTAATGGTTAGCACT CAGGACTCTGAATCCTGCGATCCGAGTTCAAATCTCGGTGGAACCT) instead of a rice glycine tRNA.

Injections were carried out by Rainbow Transgenic Flies (Camarillo, California 93012, USA) to transform flies through φC31 integrase-mediated integration into attP docker sites.

pAPIC2-ZK2-Cas9 and gRNA-marker constructs were integrated into the *attP^VK00037* site on the second chromosome, and expression vectors containing gRNAs targeting *Rab3* or *gnu* were integrated into the *attP^VK00027* site on the third chromosome. Transgenic insertions were validated by genomic PCR or sequencing.

## Identification of gRNA target sequence

gRNA target sequences for *Rab3* and *gnu* were identified as described previously [39]. Briefly, 2 gRNA prediction methods were used: sgRNA Scorer 2.0 [63] (https://crispr.med.harvard.edu) and Benchling (www.benchling.com). Candidate target sequences were those that obtained high on-target scores in both algorithms. CasFinder [48] was used to identify and reject any sequences with more than 1 target site. Two target sequences against coding exons for all splice isoforms were chosen for each targeted gene. gRNA target sequences for 2L were identified by visually scanning through pericentromeric sequences using UCSC Genome Browser (https://genome.ucsc.edu/) following principles described in the Results section. The on- and off-target scores were calculated using CRISPOR (http://crispor.tefor.net/) [64]. S2 Table lists target sequences that are used in this study and those recommended for creating gRNA-marker constructs for other chromosome arms.

## Live imaging of neurons

Live imaging was performed as previously described [62]. Briefly, animals were reared at 25˚C in density-controlled vials for between 96 and 120 hours after egg laying (to obtain third to late

third instar larvae). Larvae were mounted in glycerol, and their C4da neurons at segments A1 to A6 were imaged using a Leica SP8 confocal microscope with a 20× oil objective and a z-step size of 3.5 μm.

### Imaginal disc imaging

Imaginal disc dissections were performed as described previously [65]. Briefly, wandering third instar larvae were dissected in a small petri dish filled with cold PBS. The anterior half of the larva was inverted and the trachea and gut were removed. The sample was then transferred to 4% formaldehyde in PBS and fixed for 15 minutes at room temperature. After washing with PBS, the imaginal discs were placed in SlowFade Diamond Antifade Mountant (Thermo Fisher Scientific, Waltham, Massachusetts 02451, USA) on a glass slide. A coverslip was lightly pressed on top. Imaginal discs were imaged using a Leica SP8 confocal microscope with a 20× oil objective.

### Assays for germline mosaic analysis

To monitor mitotic recombination events resulting in germline clone generation, we performed egg laying and egg hatchability assays as detailed in [66], with the exception of using Canton-S males in place of ORP2 males as wild-type mates. Hatchability was calculated only for females that laid eggs. Females that laid no eggs were eliminated from hatchability calculations to avoid inflation of false-zero values.

### Image analysis and quantification

Counting of wing disc clones was completed in Fiji/ImageJ. Counting of neuronal clones was completed manually during the imaging process.

### Statistical analysis

Statistical analyses were performed in R. Student $t$ test was conducted for egg laying data using *Rab3* and *kni* gRNAs. For egg laying data using the 2L toolkit, we performed estimated marginal means contrasts between gRNAs and post hoc 1-sample $t$ tests using a generalized linear model with a negative binomial response. For hatchability data using the 2L toolkit, we performed estimated marginal means contrasts between proportions of hatched/nonhatched eggs for each gRNA using a generalized linear mixed-effects model with a binomial response. For all contrasts, $p$-values were corrected for multiple comparisons using the Tukey method. For the comparison of gRNAs using wing disc and neuronal clone data as well as the analysis of normalized total dendrite length, we performed Welch analysis of variance (ANOVA) followed by pairwise post hoc Welch $t$ tests. $p$-Values from the multiple post hoc Welch $t$ tests were corrected for multiple comparisons using the Bonferroni method. For hs-Cas9 data and the comparison of MAGIC and the FRT/Flp method, we performed Welch $t$ tests.

### Supporting information

**S1 Fig. Crossing scheme for germline clone induction.** (A) Crossing scheme for germline clone induction using *gRNA-Rab3*, *ovo^D1^*(2R), and *nos-Cas9*. (B) Crossing scheme for testing gRNAs for 2L in germline clone induction. The gRNAs used in this test were *gRNA(nBFP)* lines.
(TIF)

**S2 Fig. Distribution of da neuron clones using gRNA(Gal80) for 2L.** (A) Distribution of da neuron clones in each segment using gRNA(Gal80) for 40D2 and 40D4. $n$ = number of

neurons: 40D2 ($n$ = 47); 40D4 ($n$ = 52). The data underlying this Figure can be found in S1 Data. Larva drawn by G. T. K.
(TIF)

**S1 Table. Key resources.** This table lists the sources of Drosophila lines, recombinant DNAs and reagents, software, and algorithms used in this study.
(XLSX)

**S2 Table. gRNA target sequences.** This table lists target sequences on chromosome 2L that were used in this study and those recommended for creating gRNA-marker constructs for the other chromosome arms. *Doench'16 on-target score; a higher score predicts a higher efficiency. **CFD Specificity Score; a higher score corresponds to a lower off-target probability.
(XLSX)

**S1 Data. These are the numerical values of the data used to generate the plots in all figures, including the supplemental figures.**
(XLSX)

## Acknowledgments

We thank Tzumin Lee, Bloomington *Drosophila* Stock Center (https://bdsc.indiana.edu/index. html) (NIH P40OD018537), and the *Drosophila* Genetic Reference Panel (http://dgrp2.gnets. ncsu.edu/) for fly stocks; Cedric Feschotte, Andy Clark, Eric Alani, and Marcus Smolka for advice on gRNA design; Cornell CSCU consultants Stephen Parry and Erika Mudrak for advice on statistics; Michael Goldberg, John Schimenti, Erich Brunner, and Konrad Basler for critical reading and helpful suggestions on the manuscript.

## Author Contributions

**Conceptualization:** Sarah E. Allen, Gabriel T. Koreman, Ankita Sarkar, Bei Wang, Mariana F. Wolfner, Chun Han.

**Data curation:** Sarah E. Allen, Gabriel T. Koreman, Ankita Sarkar, Bei Wang.

**Formal analysis:** Sarah E. Allen, Gabriel T. Koreman, Mariana F. Wolfner, Chun Han.

**Funding acquisition:** Mariana F. Wolfner, Chun Han.

**Investigation:** Sarah E. Allen, Gabriel T. Koreman, Ankita Sarkar, Bei Wang.

**Methodology:** Sarah E. Allen, Gabriel T. Koreman, Ankita Sarkar, Bei Wang.

**Supervision:** Mariana F. Wolfner, Chun Han.

**Validation:** Sarah E. Allen, Gabriel T. Koreman, Ankita Sarkar, Bei Wang, Mariana F. Wolfner, Chun Han.

**Visualization:** Sarah E. Allen, Gabriel T. Koreman, Ankita Sarkar, Bei Wang, Mariana F. Wolfner, Chun Han.

**Writing – original draft:** Sarah E. Allen, Gabriel T. Koreman, Ankita Sarkar, Mariana F. Wolfner, Chun Han.

**Writing – review & editing:** Sarah E. Allen, Gabriel T. Koreman, Mariana F. Wolfner, Chun Han.

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
