## [Editor Report · Decision Letter 0]

1 Jul 2020

Dear Mariana, 

Thank you for submitting your manuscript entitled "MAGIC: Mosaic Analysis by gRNA-Induced Crossing-over" for consideration as a Methods and Resources by PLOS Biology.

Your manuscript has now been evaluated by the PLOS Biology editorial staff, as well as by an academic editor with relevant expertise, and I'm writing to let you know that we would like to send your submission out for external peer review.

Please re-submit your manuscript within two working days, i.e. by Jul 03 2020 11:59PM.

Best wishes,

Roli

Senior Editor

PLOS Biology

---

## [Decision Letter · Decision Letter 1]

5 Aug 2020

Dear Mariana,

Many thanks for submitting your manuscript "MAGIC: Mosaic Analysis by gRNA-Induced Crossing-over" for consideration as a Methods and Resources at PLOS Biology. Your manuscript has been evaluated by the PLOS Biology editors, an Academic Editor with relevant expertise, and by two independent reviewers.

You'll see that both reviewers are broadly positive about your study, but each raises a number of concerns that will need to be addressed both textually and with new experimental data. In light of the reviews (below), we will not be able to accept the current version of the manuscript, but we would welcome re-submission of a much-revised version that takes into account the reviewers' comments. We cannot make any decision about publication until we have seen the revised manuscript and your response to the reviewers' comments. Your revised manuscript is also likely to be sent for further evaluation by the reviewers.

We expect to receive your revised manuscript within 3 months. 

**IMPORTANT - SUBMITTING YOUR REVISION**

*Re-submission Checklist*

*Published Peer Review*

*PLOS Data Policy*

*Blot and Gel Data Policy*

Best wishes,

Roli

Senior Editor,

rroberts@plos.org,

PLOS Biology

REVIEWERS' COMMENTS:

Reviewer #1:

Allen and colleagues describe a novel CRISPR/Cas9-based technique in Drosophila enabling the generation of genetic mosaics through interchromosomal recombination. The authors validate their approach in a number of somatic tissues and the female germline. The authors also provide a tool set for the genetic mosaic-based functional study of candidate genes on chromosome arm 2L. Lastly, the authors cross their transgenic lines to fly lines with wild-derived genomes and provide proof-of-principle.

Overall the authors present a potentially useful method for the study of candidate genes in genetic mosaic flies. Currently, the manuscript presents mainly proof-of-principle and the resource is still limited. While the data presentation is sound, the writing of the manuscript could be improved. At many passages the writing is not precise. The reader gets the impression that the authors slightly oversell their method, especially since CRISPR/Cas9-based clonal analysis is not new. Likewise, interchromosomal recombination to generate genetic mosaics is (as the authors nicely document in the Introduction) not a new approach. The combination of the two represents certainly an advance, but there are other methods that achieve the same result and I think the authors should pay special attention to not discredit established FRT/Flp-based methods. The authors should also tone down certain claims. Below I point out more specific points that require attention:

1. In the Abstract, line 21 the authors state: '…and can be applied in any organism that…' The authors should say '… can in principle be applied…' They do not show any example where their method has been validated in another organism.

2. In the Introduction, line 31 authors state: …Mosaic (also called clonal) analysis … this is simply wrong. Mosaic and clonal are two different things and the authors should please pay attention that the writing is precise. This is actually important throughout the entire manuscript. The authors at most places talk about 'clonal analysis' but how often do the authors really know that the labeled/mutated cells derive from one individual progenitor stem cell? In a clone, all cells are lineally related and derive from a single stem cell. The authors should, especially in such a methods paper, not confuse clonal with mosaic. Cells in a genetic mosaic can be a single clone of mutated/labeled cells in a tissue/animal, but most often mosaic simply means that labeled/mutated cells are present within a genetically distinct background. The cells in a mosaic do not need to be clonally-related. In fact the authors do not use inducible Cas9. Thus they will never know when the DSB was induced and interchromosomal recombination happened. As a consequence they also will not know if more than one event happened and in more than one progenitor stem cell.

3. Line 82 - the authors state: '…, a 50% chance exists for identical distal chromosome segments to sort into the same daughter cell, generating …' The authors should please precise their writing and explain the entire spectrum of segregation possibilities in more detail. If recombination happens in G2 phase of the cell cycle, there are two segregation possibilities, G2-X (recombinant chromosomes segregate away from each other) and G2-Z (recombinant chromosomes 'sort' together into the same cell). The authors should elaborate for the non-specialist reader and clarify the schematic in Figure 1.

4. The authors state on line 119 - '…label clones homozygous … either negatively or positively…'. How can something be negatively labeled. Please be precise the writing.

5. Line 129 - '… as to avoid off-target effects…'. Can the authors estimate the probability for off target effects for the gRNA they used? A 'unique' target site does not ensure that no off-target effects occur. More generally, off-target effects cannot be completely excluded when using MAGIC and the authors should discuss this caveat in the Discussion.

6. The authors state on line 146: '… suggests that an efficient gRNA construct for one tissue will likely perform well in other tissues also…'. Please remove such speculation or show the data with quantitative assessment. Also in the discussion line 195/6.

7. The authors repetitively state that '… analogous toolkits could easily be made for any other chromosome arm…' (e.g. line 148/9 and 177/8). Well, if it is so easy, why did the authors not do it and provide a more complete resource?

8. Line 169/170 - '…demonstrating the potential of MAGIC for clonal analysis of the function of natural alleles residing…'. The authors did not do any functional analysis of natural alleles. Please tone down the claims or at least discuss such issues in a more balanced manner.

9. The entire Discussion aims to sell the method, which in principle is fine. However, the entire Discussion should be written in a more balanced manner. Please do not discredit previous work based on FRT/Flp-mediated recombination. There are thousands of studies that exploit such technique and even once MAGIC is available for the community, many people will continue using FRT/Flp systems for functional gene analysis. In fact, FRT sites have been inserted and reliably used on (almost) all chromosome arms. MAGIC currently is only enabling functional gene analysis on 2L.

Reviewer #2:

[identifies himself as Liqun Luo]

In this manuscript, the authors described a new method for mosaic analysis in Drosophila. Their method utilizes CRISPR/Cas9-induced double-strand breaks (DSBs) to induce mitotic recombination and thus can randomly produce homozygous clones in a heterozygous animal.

MAGIC offers two advantages. (1) The genetics is simpler compared to conventional approaches. (2) Because MAGIC does not require FRT sites, it will allow mosaic analysis for genes that cannot be analyzed by traditional approaches due to the position of existing FRTs. Thus, this new tool has potential to make mosaic analysis more doable and efficient, and even beyond the use in Drosophila.

I would be enthusiastic in supporting the publication of this manuscript if the authors can address the following issues in a revised manuscript:

1. The authors should compare the efficiency of gRNA-induced crossing-over with the efficiency of recombination mediated by FRT sites. This information will be critical for other researchers to determine if MAGIC is suitable for their study. We worry the efficiency could be markedly lower in MAGIC because DSB repair by non-homologous end joining (NHEJ) happens more frequently than homology-directed repair (HDR). In addition, when NHEJ occurs, it can mutate the gRNA target sites which prevents subsequent Cas9 cutting. 

2. The author should demonstrate a temporally inducible way of making clones—such as using a heat shock promoter to drive Cas9. This is one of the most widely used ways to induce clones in the field. This is especially important in light of the caveat the authors raised in their discussion: "For the cell type in question, an ideal Cas9 should be expressed in the precursor cells, as too early expression can mutate gRNA target sites prematurely and too late expression will lead to unproductive DSBs."

3. Mosaic analysis technique papers in Drosophila have typically included resources that allow researchers to use the tools right away, rather than having to create the tools AND apply the tools. This will speed up the adoption of new techniques. The authors have produced tools for analysis of genes located on 2L, which covers only 20% of the genome. The paper will be greatly improved if the authors can also provide tools for other chromosomal arms. These will also serve to further validate the generality of the approach. I understand that this is a substantial amount of work (creating new transgenes) in particular during the pandemic, so I will leave it up to the journal editors to decide whether it is an option or a requirement. 

Minor issues:

4. The statement on line 82 (a 50% chance…) is incorrect. G2-X (Fig. 1A top) and G2-Z (Fig. 1A, bottom) segregations are known to be unequal. There is a literature on this in Drosophila and in other organisms. 

5. The illustration of MAGIC events should be kept consistent throughout the paper. Simplifying the sister chromatids to just one line can cause some confusions regarding when MAGIC event occurs (Figure 1B, 1F, 2A, 2B).

6. In the legend for Figure 2C, what the grey boxes represent should be stated clearly.

7. In figure 3, the total number of clones and the penetrance of the phenotype should be shown.

8. In line 181, "genetic modification" should be stated more explicitly as 'the requirement of FRT sites on the homologous chromosomes'.

9. The authors stated that the ability to use MAGIC on DGRP wild-derived strains is one of its major advantages. However, we hoped that there can be more explanation for it. Specifically, the statement "it has been difficult to investigate the effect of homozygosity for alleles within these strains without being able to use available genetic tools" (line 212-213) is a bit confusing. Our understanding is that MAGIC cannot be directed crossed to those flies—either Cas9 or the MAGIC gRNA has to be combined with the allele first before the flies that produce mitotic recombination can be put together—to avoid keeping Cas9 and gRNA in the same fly across a generation. If this is the case, wouldn't MAGIC suffer the same constraint as other approaches?

10. Drosophila has an unusual property that homologous chromosomes pair even in mitotic cell cycles, facilitating mitotic recombination. Most organisms do not have this property so it is unclear whether MAGIC would work. In their enthusiasm to state that MAGIC can be applied to other organisms, the authors should add this caveat.

---

## [Decision Letter · Decision Letter 2]

1 Dec 2020

Dear Mariana,

Thank you for submitting your revised Methods and Resources paper entitled "MAGIC: Mosaic Analysis by gRNA-Induced Crossing-over" for publication in PLOS Biology. I have now obtained advice from the original reviewers and have discussed their comments with the Academic Editor. 

Based on the reviews, we will probably accept this manuscript for publication, assuming that you will modify the manuscript to address the remaining points raised by the reviewers. 

IMPORTANT:

a) Please attend to the remaining requests from reviewer #2. Regarding this reviewer's point 1, the Academic Editor says that this can be addressed "by incorporating some additional possibilities in the discussion or by toning down the main statement."

b) Many thanks for providing the data underlying the Figures in your supplementary S1_Data file. Please could you cite this in all relevant main and supplementary Figure legends? e.g. "The data underlying this Figure can be found in S1 Data."

c) We wonder if you could consider something more explicit for your title, to make your method's applications more evident? Perhaps "Using gRNA and CRISPR technology to facilitate mosaic analysis on a wide range of model organisms" or some such.

We expect to receive your revised manuscript within two weeks. Your revisions should address the specific points made by each reviewer. In addition to the remaining revisions and before we will be able to formally accept your manuscript and consider it "in press", we also need to ensure that your article conforms to our guidelines. A member of our team will be in touch shortly with a set of requests. As we can't proceed until these requirements are met, your swift response will help prevent delays to publication.

- a cover letter that should detail your responses to any editorial requests, if applicable

*Copyediting*

*Published Peer Review History*

*Early Version*

Best wishes,

Roli

Senior Editor,

rroberts@plos.org,

PLOS Biology

REVIEWERS' COMMENTS:

Reviewer #1:

The authors have addressed all concerns sufficiently. The newly added data and rewriting improved the manuscript.

Reviewer #2:

[identifies himself as Liqun Luo]

In the revised manuscript, the authors provided additional data to demonstrate the compatibility of MAGIC with heat shock and compared the efficiency of MAGIC with traditional FLP/FRT method. They have also incorporated textual changes that addressed most of our previous concerns. However, we hoped that the authors can address following points regarding their new data:

(1) The results of comparing MAGIC with FLP/FRT is not completely satisfying. Firstly, the authors only compared one MAGIC construct (gRNA-40D2) with one FRT site (FRT40A). Without more thorough analysis using more examples, it is a bit of a stretch to generalize this observation to the conclusion that "Flp transgenes were much less efficient in generating clones than their Cas9 counterparts" (line 185-186). Secondly, no explanation is provided for why gRNA-40D2 is about 10-fold more efficient than FRT40A in the text. Thirdly, while zk-Cas9 generated fewer but larger clones and hh-Cas9 generated more but smaller clones, there is no significant differences between the clones generated by zk-FLP and hh-FLP. The authors should provide explanation for this.

Minor issues:

(2) There is inconsistency between text and figure legend for which gRNA is used for MAGIC and FLP/FRT comparison experiment. In line 182, it says "gRNA-40A (nBFP)". But in legend line 643 it says "gRNA-40D2 (nBFP)".

(3) In Figure 3D, shouldn't the heterozygous mother cell have blue and red mixture color instead of grey?

(4) Based on the image examples, the branching number is also changed in Syx5 mutant comparing to wild type. Is this a common phenomenon for all Syx5 mutant?

---

## [Editor Report · Decision Letter 3]

4 Jan 2021

Dear Dr. Wolfner,

I am writing concerning your manuscript submitted to PLOS Biology, entitled “Versatile CRISPR/Cas9-mediated mosaic analysis by gRNA- induced crossing -over for unmodified genomes.”

We have now completed our final technical checks and have approved your submission for publication. You will shortly receive a letter of formal acceptance from the editor.

Kind regards,

PLOS Biology